# Persistent inequality in economically optimal climate policies

Paolo Gazzotti [1✉], Johannes Emmerling [2], Giacomo Marangoni[1,2], Andrea Castelletti [3], Kaj-Ivar van der Wijst [4], Andries Hof [4] & Massimo Tavoni [1,2✉]

Benefit-cost analyses of climate policies by integrated assessment models have generated conflicting assessments. Two critical issues affecting social welfare are regional heterogeneity and inequality. These have only partly been accounted for in existing frameworks. Here, we present a benefit-cost model with more than 50 regions, calibrated upon emissions and mitigation cost data from detailed-process IAMs, and featuring country-level economic damages. We compare countries' self-interested and cooperative behaviour under a range of assumptions about socioeconomic development, climate impacts, and preferences over time and inequality. Results indicate that without international cooperation, global temperature rises, though less than in commonly-used reference scenarios. Cooperation stabilizes temperature within the Paris goals (1.80°C [1.53°C–2.31°C] in 2100). Nevertheless, economic inequality persists: the ratio between top and bottom income deciles is 117% higher than without climate change impacts, even for economically optimal pathways.

[1] Department of Management, Economics and Industrial Engineering, Politecnico di Milano, Milan, Italy. [2] RFF-CMCC European Institute on Economics and the Environment (EIEE), Centro Euro-Mediterraneo sui Cambiamenti Climatici, Milan, Italy. [3] Department of Electronics, Information, and Bioengineering, Politecnico di Milano, Milan, Italy. [4] PBL Netherlands Environmental Assessment Agency, The Hague, the Netherlands. ✉email: paolo.gazzotti@polimi.it; massimo.tavoni@polimi.it

Traditional benefit-cost analysis based on neoclassical Integrated Assessment Models (IAM) has often found limiting global warming to well-below 2 °C to be economically inefficient. In its standard set-up, the Dynamic Integrated Climate-Economy (DICE) model, the best-known benefit-cost IAM, suggests the economically optimal temperature to be 3.5 °C in 2100, and to peak above 4 °C amid next century[1]. This largely exceeds the UNFCCC Paris Agreement target[2] and is close to what foreseen by detailed-process IAMs under lenient climate policies.

The DICE model has thus been subjected to several criticisms. Disputed points have been the choice of normative parameters such as the social rate of time preference and the inter-temporal elasticity of substitution[3–5], the specification of impact functions[6,7], the model's climate module[8], and the mitigation cost structure[9,10]. Thanks to being open-source and easily accessible, many studies have shown the model sensitivity to these factors.

Recently, few papers[11–14] have amended DICE to consider the latest climate and economics science, including empirically-derived climate impact functions. The re-calibrated models have shown that the Paris agreement targets might be economically optimal under standard benefit-cost analysis. However, these findings rely on a single-region model, with no consideration of regional heterogeneity and inequality in the costs and benefits of climate action. This is a major limitation: one key finding of the empirical evidence about climate economic impacts is their high heterogeneity across countries[15–18]. Furthermore, emission reduction opportunities as estimated by detailed-process IAMs and reviewed by the IPCC vary significantly across countries[19]. Finally, aggregated benefit-cost analyses hide key sources of inequality, and these are consequently not accounted for in their welfare frameworks. The evidence from the benefit-cost assessments of heightened climate change impact doesn't yet account for regional heterogeneity. This prevents a comparison between cooperative and self-interested scenarios, accounting for preferences over equality, and fully capturing economic convergence dynamics and technological progress. Most importantly, it obscures inequalities across countries and leaves them out entirely from the optimization.

The benefit-cost literature has examined heterogeneity and inequality before. Ricke et al.[20] provide a comprehensive analysis on the social cost of carbon (SCC) at the country-level resolution, while Taconet et al.[21] evidence how climate change affects inequality between countries under a large variety of scenarios. However, both do not perform any optimal evaluation. Benefit-cost IAMs such as RICE[22,23], AD-RICE[24], PAGE[25], FUND[26], CWS[27], MICA[28], C3IAM[29], and STACO[30], disaggregate the global economy in up to 6–16 macro-regions (see also the review by Weyant[31]). This resolution allows only to partially capture the variation in the costs and benefits of climate action, and none of these models accounts for the latest climate and economic evidence. While other models, including Computable General Equilibrium (CGE) models such as in[32], provide sectoral and regional detail, their policy questions are in most cases evaluating prescribed policy pathways rather than inter-temporal optimization. Thus, albeit the topic has been addressed before, the proposed framework's improved granularity, calibration, and welfare specification represent novel steps.

We show that it is possible and advisable to go beyond traditional benefit-cost analysis centered on economic efficiency and to include heterogeneity and inequality as one of the key components of welfare. To do so, we have extended, regionalized and re-calibrated the DICE Integrated Assessment model to more than 50 independently modeled countries or regions, taking into account the latest evidence and data and expanding the social welfare function to include inequality aversion (see next section and Methods). The new model highlights international cooperation's relevance for achieving climate targets compliant with the Paris agreement. We also show and quantify how climate change increases global income inequalities even under welfare-maximizing policies and different socioeconomic pathways.

## Results

**RICE50+ model and scenarios.** The modeling framework considers 57 independent regions (or countries, see Supplementary Information for the full list). The dynamics of economic growth, greenhouse gases (GHG) emissions, emissions mitigation costs, and economic impacts due to climate change follow the well-known integrated assessment model DICE (the latest version of DICE-2016R2, used in[1]). The regional representation is consistent with the finest granularity at which data, especially marginal abatement costs curves (MACC), is available. Socioeconomic drivers, including population and economics growth at the country level, come from the five Shared Socioeconomic Pathways (SSP)[33,34]. Therefore, the model spans over five coherent alternative future socioeconomic development pathways.

The climate is modeled based on the original three-layered carbon-cyle structure, with exchange coefficients recalibrated to match the MAGICC6 model emulation[35]. Radiative forcing and atmospheric temperature increase are also evaluated at the regional level through statistical downscaling calibrated on the CMIP5 database[36].

This framework allows us to directly introduce empirically estimated climate impact functions at the country level, without the need to resort to aggregate-response fitting as in Glanemann et al.[11]. Climate, therefore, influences GDP growth according to local temperature variations. We use empirically-estimated non-linear impact functions which relate temperature increase to economic growth[15]. We consider all four major empirical specifications of Burke-Hsiang-Miguel (BHM). These include different time lags -capturing short-run (SR) and long-run (LR) impacts- and the extent to which rich and poor countries' income differentiation is accounted for. They also implicitly allow for different historical adaptation to climates. Furthermore, we carry out robustness analysis with alternative empirical impact function studies (Dell et al.[17], Kahn et al.[18]).

Emissions and marginal abatement cost curves are calibrated on multiple sources. For the near future (2025–2040), we use Enerdata-EnerFuture curves, based on the detailed-process-based model POLES[37], an energy sector model jointly developed with the European Commission. For the rest of the century, we use the information on emissions and abatement potential from detailed-process IAMs reviewed in the IPCC SR1.5[19]. In the very long run (post-2100), model assumptions converge to DICE trends. Constraints on emission reduction rates, due to the energy system inertia, and negative emissions availability follow[12].

Regions maximize inter-temporal welfare. When acting in self-interest, countries act non-cooperatively and optimize their mitigation strategy taking others' behavior as given. Thus only own country climate impacts are optimized upon. The Nash equilibrium is found through an iterative algorithm. On the other hand, the cooperative setting implies a global social planner who maximizes a social welfare function that aggregates all regions' welfare. The implicit normativity of IAMs often doesn't include redistributive preferences. In contrast, our welfare specification disentangles inequality-aversion and inter-temporal inequity aversion. This allows capturing the essential issues of inter-temporal and spatial inequality (see Anthoff and Emmerling[38] and Atkinson et al.[39] and the Methods section).

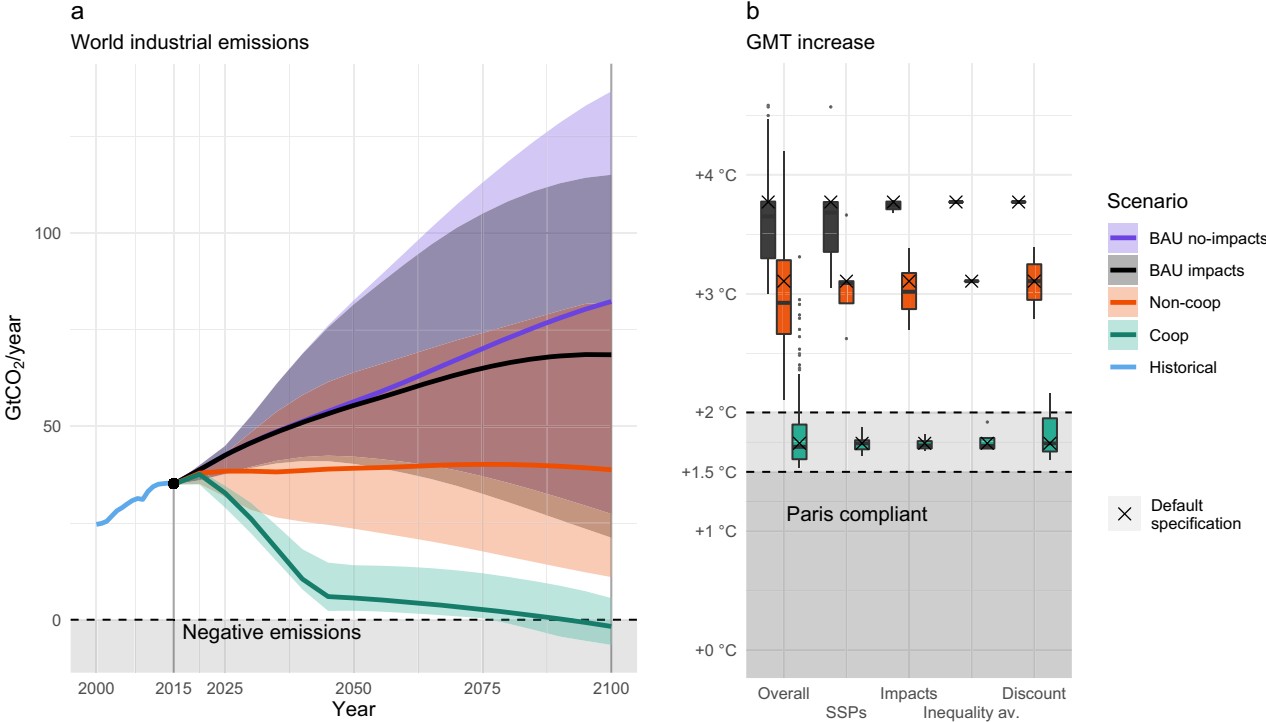

**Fig. 1 Optimal world-aggregated model outcomes. a** Global $CO_2$ emissions pathways. Uncertainty ranges include all SSPs and climate change impacts. **b** Global Mean Temperature (GMT) increase in 2100, and decomposed uncertainties ranges. *Overall* accounts for all the uncertainties. Other factors show temperature variability due to every single driver (with others fixed at their default level). Default specification (SSP2, BHM-SR impacts, utility discount rate of 1.5% and $\gamma = 0.5$) values are highlighted with a cross marker.

The geographical resolution, for both benefits and costs, and the expanded welfare function, allow us to explore the key sources of heterogeneity and inequality. These include the degree of cooperation (non-cooperative, cooperative), socioeconomic trends (SSP1 to SSP5, with SSP2 as default), four climate impact specifications (BHM-SR as default), and inequality aversion ($\gamma$). We span from inequality neutrality ($\gamma = 0$) to high inequality aversion ($\gamma = 2$), covering the full range suggested by Atkinson and Brandolini[40]. We choose $\gamma = 0.5$ as our default value, in accordance with several sources[41,42]. We also run alternative values for the utility discount rate $\rho$, varying over the interval 0.1–3% (1.5% as default). Unless otherwise stated, these values are adopted as default.

**Global outcomes**. Figure 1 summarises the world-aggregated model outcomes. The Business-As-Usual scenarios (*BAU no-impacts*), which assume no mitigation, project typically increasing emissions trends (Fig. 1a). When climate impacts are factored in -a mechanism absent in the usual reference scenarios and yet relevant[43]- the lowered economic growth rate slightly reduces emissions (*BAU impacts*). This leads to a mean temperature increase at the end of the century of +3.65 °C [2.99–4.49 °C, 10th-90th percentile range] (Fig. 1b). However, an adequate counter-factual scenario is when countries react to climate impacts based on their pure self-interest (here labeled as *Non-coop*). The non-cooperative scenario is characterized by relatively flat emissions, with an average 2100 temperature increase of +3 °C [2.10–4.19 °C] over pre-industrial levels. This result highlights the importance of an appropriate baseline, and corroborates recent criticisms of counterfactual scenarios having implausibly high emissions[44]. A proper accounting of climate economic feedback generates significantly lower emissions and forcing than in the original SSPs[34]. Figure 1 also shows that if countries cooperated for the sake of the global good (*Coop*), fast emissions reductions would be optimal. Global carbon neutrality would be approached by mid-century. In

most cases, these cooperative scenarios have a temperature increase below +2 °C [1.80, 1.53–2.31 °C]. These results confirm the recent DICE-based findings by Glanemann et al.[11] and Hänsel et al.[12] in a regional setting. They also show that the Paris agreement's stricter interpretation of 1.5 °C is not cost-optimal[14]. In terms of the uncertainty characterizing these global outcomes, Fig. 1b summarizes the distribution of 2100 temperature increase, disentangling the contributions of every driver (with the others kept at their default level). Some sources of uncertainty, such as the normative decision of discount rate, equally affect both cooperative and non-cooperative outcomes. Others, such as socioeconomic pathways and climate impacts, have differentiated consequences, with non-cooperation showing wider outcome ranges. See also Supplementary Fig. 10 for different uncertainty ranges in world-aggregated emissions.

**Major economies and NDCs comparison**. Figure 2 compares mitigation efforts and costs among selected major economies. Both cooperative and non-cooperative scenarios are reported for the policy-relevant time-frames of 2030 and 2050. In the absence of cooperation, emission reductions vary significantly across countries but are positive for most of them. Notably, large countries with relatively low mitigation costs and high expected climate impacts (India, followed by China and the U.S.) mitigate emissions significantly out of pure self-interest. These countries cut CO2 between 20% to up to 75% of BAU emissions in 2050 (Fig. 2c). Under full cooperation, all regions reduce emissions close to the maximum potential, except Sub-Saharan Africa which starts at very low emissions per-capita level. This is consistent with mitigation efforts in the major world economies aiming at climate stabilization[45]. We compare the regional emissions with the pledges made by countries in the Paris Accord for 2030. National Determined Contributions (NDCs), as estimated by Hof et al.[46] and den Elzen et al.[47] and reported in Fig. 2a, are closer to

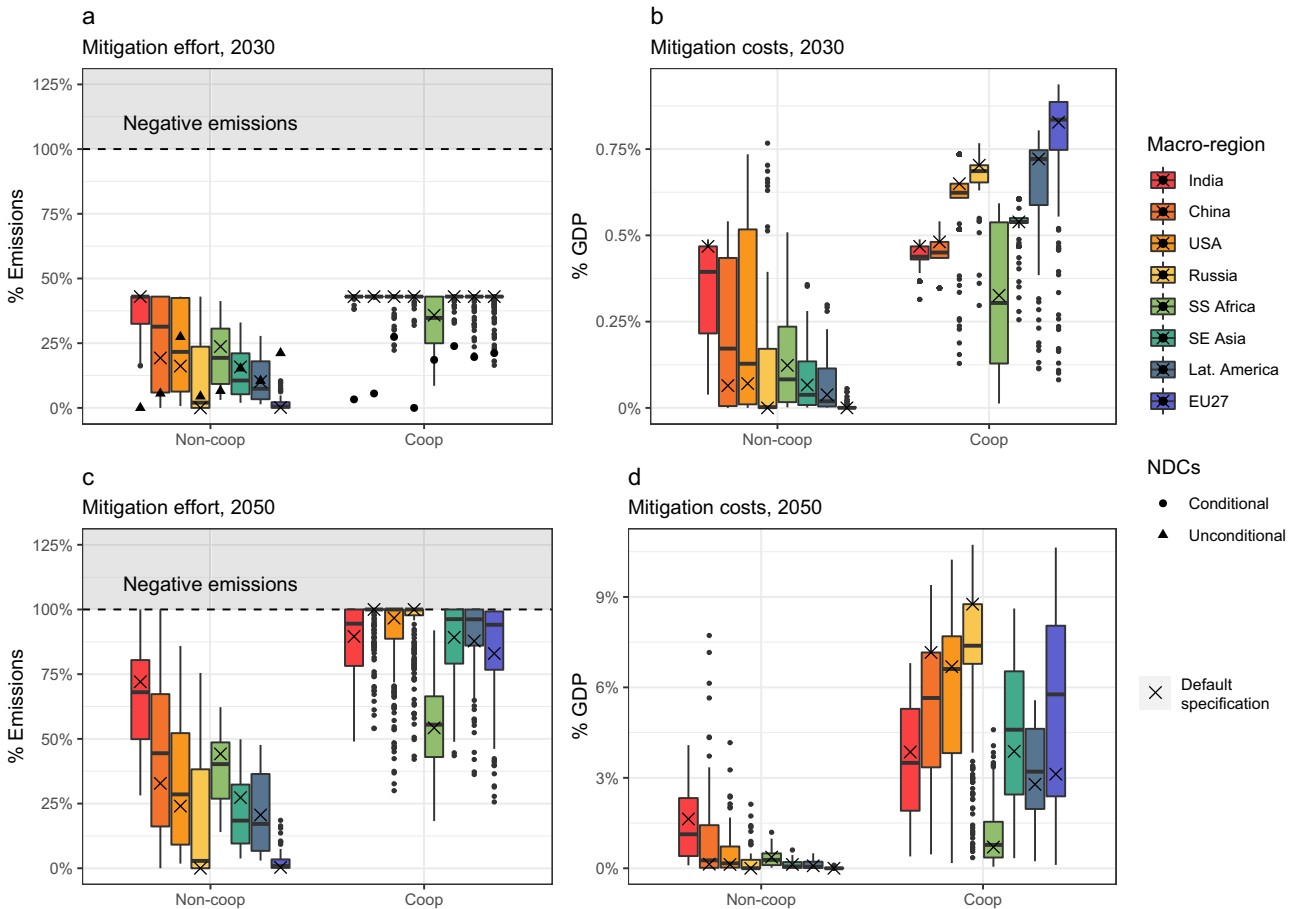

**Fig. 2 Major economies mitigation efforts and costs for cooperative and non-cooperative scenarios. a**, **b** Distributions for year 2030 and NDCs pledges. **c**, **d** Distributions for year 2050. Mitigation efforts and costs are reported as the percentage of emissions reduced and percentage of GDP over BAU no-impacts reference. Macro-regions show the aggregated level from finer geographical-resolution results. Boxplot ranges include all scenarios explored. Default specification (SSP2, BHM-SR impacts, $\rho = 1.5\%$, and $\gamma = 0.5$) values are also highlighted.

non-cooperative ranges. Notable exceptions are India (conditional NDC target estimated around 10% of reductions, lower than its optimal self-interest effort) and EU27 (aiming at least 25% of reductions, with even higher ambitious decarbonization programs under consideration), which are closer to – but still lower than – cooperative levels. On the other hand, cooperation demands higher ambition than the NDCs for all regions, supporting a ratcheting up of the current pledges[48]. Emission reductions translate into costs (Fig. 2b, d), depending on the intensity of the mitigation effort and regional abatement opportunities. In 2030, GDP losses are negligible, but they increase to up to 10% in 2050 for the most exposed regions (i.e., Russia). The heterogeneity of mitigation costs related to, among other factors, the carbon intensity of the economy has been documented with detailed-process IAMs[45]. Additional details for important regions such as the Middle East and individual countries are shown in Supplementary Fig. 1.

**Persistent inequality**. Turning to equity considerations, Fig. 3 shows the distributional effects of costs and impacts for globally cooperative scenarios. Mitigation costs are spread relatively uniformly across countries' income levels (for the default value of inequality aversion). However, climate change impacts are highly regressive. The poorest countries (representing the largest share of the world population, as indicated by the bubble size) face dramatic economic losses, exceeding 20% of GDP. Indicatively, the

economic impacts increase by 11 percentage points for every 10,000 Dollars reduction in per-capita GDP. Note that this happens despite the strong collective effort to reduce emissions, which, as we have seen, allows to keep temperature below 2 °C.

As reported in Supplementary Fig. 4, different inequality aversion parameters lead to markedly different regional mitigation efforts. For every 10,000 Dollar lower GDP level, mitigation costs ranges from an increase by 0.2 ($\gamma = 0$) to a decrease by 0.6 ($\gamma = 1.45$) GDP percentage points. Supplementary Fig. 8c relates the choice of inequality aversion to per-capita emissions, thus connecting the welfare representation of preferences over equity to the debate about burden-sharing. Variation in inequality preferences leads to vastly different per-capita emissions, consistent with commonly discussed burden-sharing principles such as equality per-capita. Yet, globally aggregated emissions and resulting temperature are relatively unchanged, as shown in Fig. 1b. Thus, climate damages remain highly regressive independently of equity preferences; these mostly affect mitigation efforts and costs.

This analysis is robust also across different climate impact specifications (see Supplementary Fig. 3). These results show that global aggregated impacts may be misleading, given the wide range of expected impacts (between −50% and +50%), even in the most optimistic cooperative scenario (roughly compliant with the Paris target).

Further evidence on the persistent inequalities in economically optimal pathways is reported in Fig. 4. It highlights how climate change stretches the income distribution between countries. The

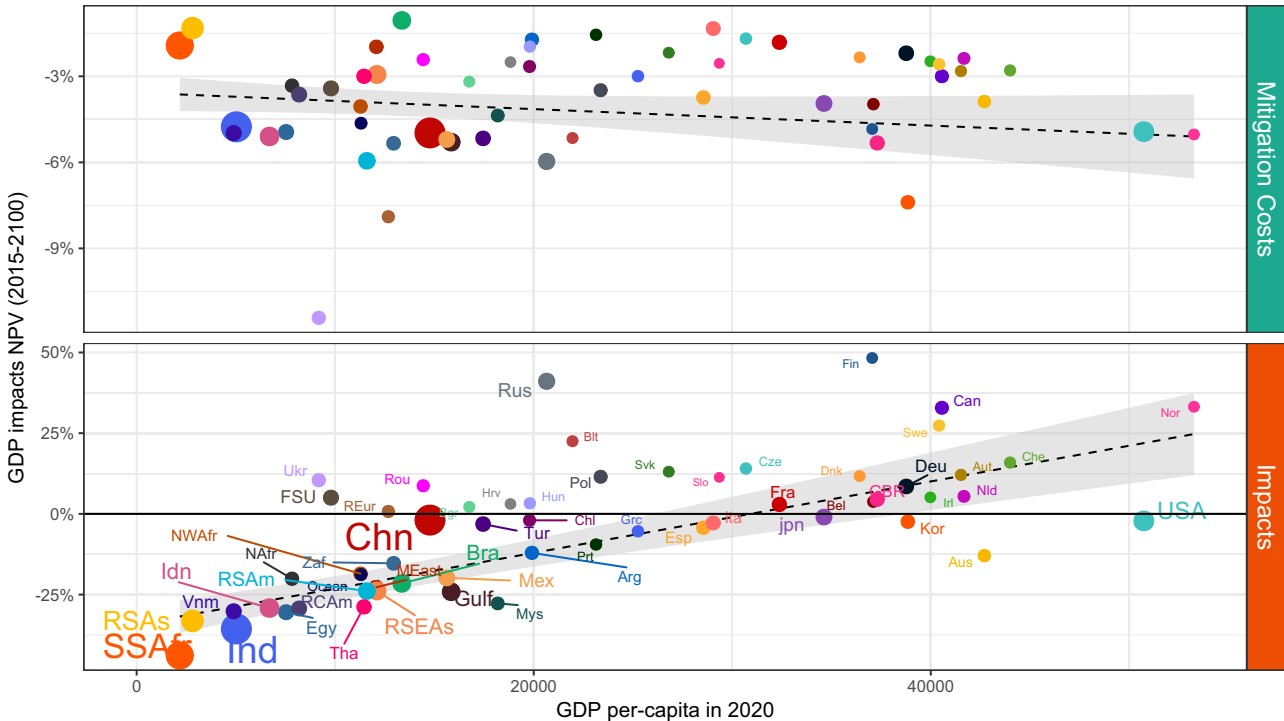

**Fig. 3 GDP net-present-value impacts and costs against current GDP per-capita.** NPV is evaluated for interval 2015–2100, with world-average Ramsey yearly discount rate. Reported scenario is cooperative, SSP2, under BHM-SR impact function and intermediate levels for inequality aversion ($\gamma = 0.5$) and utility discount rate ($\rho = 1.5\%$).

shift towards poorer levels affects most of the world population, while a smaller number of countries gain. The reason for this is the non-linearity of climate impacts with temperature. Despite being affected by lower local temperature increases than higher-latitude countries, being already far from the optimal temperature (estimated around 13 °C by Burke et al.[15]) results in heightened damages for warmer regions (see Fig. 5). The consequence is that 2.3 and 1.6 billion of African and Indian citizens respectively see their income drop by up to 60%. Climate-driven inequalities are significant even under globally optimal scenarios. Due to climate change, the ratio between top and bottom income deciles (90:10 ratio) and quintiles (80:20 ratio) increases by 117% and 63% respectively in the cooperative scenario (panel b, Fig. 4). This adds to warming already observed today, which has led to an increase of the top-bottom income decile ratio of 25%[16] (although calculated at a finer regional scale). An extra degree of warming has a more-than-proportional effect on global inequality due to the non-linearity of climate economic impacts (see Supplementary Fig. 9). The inequality ratio improves over time (panel c, Fig. 4) thanks to global economic convergence; growing uncertainties however accompany this projection.

The relevance of the underlying socioeconomic development is shown in Supplementary Fig. 6. Inequality trends are coherent with the socioeconomic storylines and assumptions about economic catch-up between developing and industrialized countries. Climate increases economic inequality consistently across SSPs. Under SSP4, a pathway characterized by persistent inequalities, the inequality indexes at the end of the century are higher than today (by 40% for the 90:10 ratio). In SSP1, characterized by a low emission and economically equitable outlook, inequality improves over today (by 79% for 90:10); but remains below where it would have been without climate change (90% reduction). Failure to globally cooperate on emission reductions increases inequality, by more than doubling the income ratios. Results are robust across different impact specifications (as shown in

Supplementary Fig. 7) and alternative levels of inequality aversion (as shown in Supplementary Fig. 8). Compared to no-climate-change BAU, only under BHM-LR assumption, which projects higher losses and affects all the countries, we observe cooperation leading to small improvements (about −5.3%); in all other cases, it worsens more than ten times in magnitude. Lower tolerance for inequality in the welfare specification improves inequality only marginally, as reported in Supplementary Fig. 8.

Taken together, these results project climate-induced inequalities both under self-interested and cooperative behavior and preferences for equality. The main determinant of this result is the climate impact-functions and their non-linear response to local temperature increases, which determine long-lasting economic growth reductions. Consequently, and together with the inertia in the mitigation ramp-up speed, emission reductions start bearing a visible effect on temperatures only around 2050 and beyond (see Supplementary Fig. 5b). The combined effect of historical emissions and those occurring in the few coming decades is sufficient to irreversibly increase inequality between countries and lead to strong climate impacts even if global cooperation is achieved, as shown in Fig. 5. This result is robust to alternative impact specifications (as shown in Supplementary Fig. 2). Moreover, if full cooperation is delayed to 2030, a more realistic outcome consistent with the current pledges, inequality would further increase (from an average of +117 % to +148% for the 90:10 ratio over baseline values, see Supplementary Fig. 5). Note that we did not model within-country inequality in our analysis. Adding this additional level of disaggregation, as indicated, e.g., by Dennig et al.[49] and Hsiang et al.[50], would increase global inequalities.

## Discussion
The regional-detailed and welfare-expanded benefit-cost analyses provide a broader, more burdened, perspective of the evolving

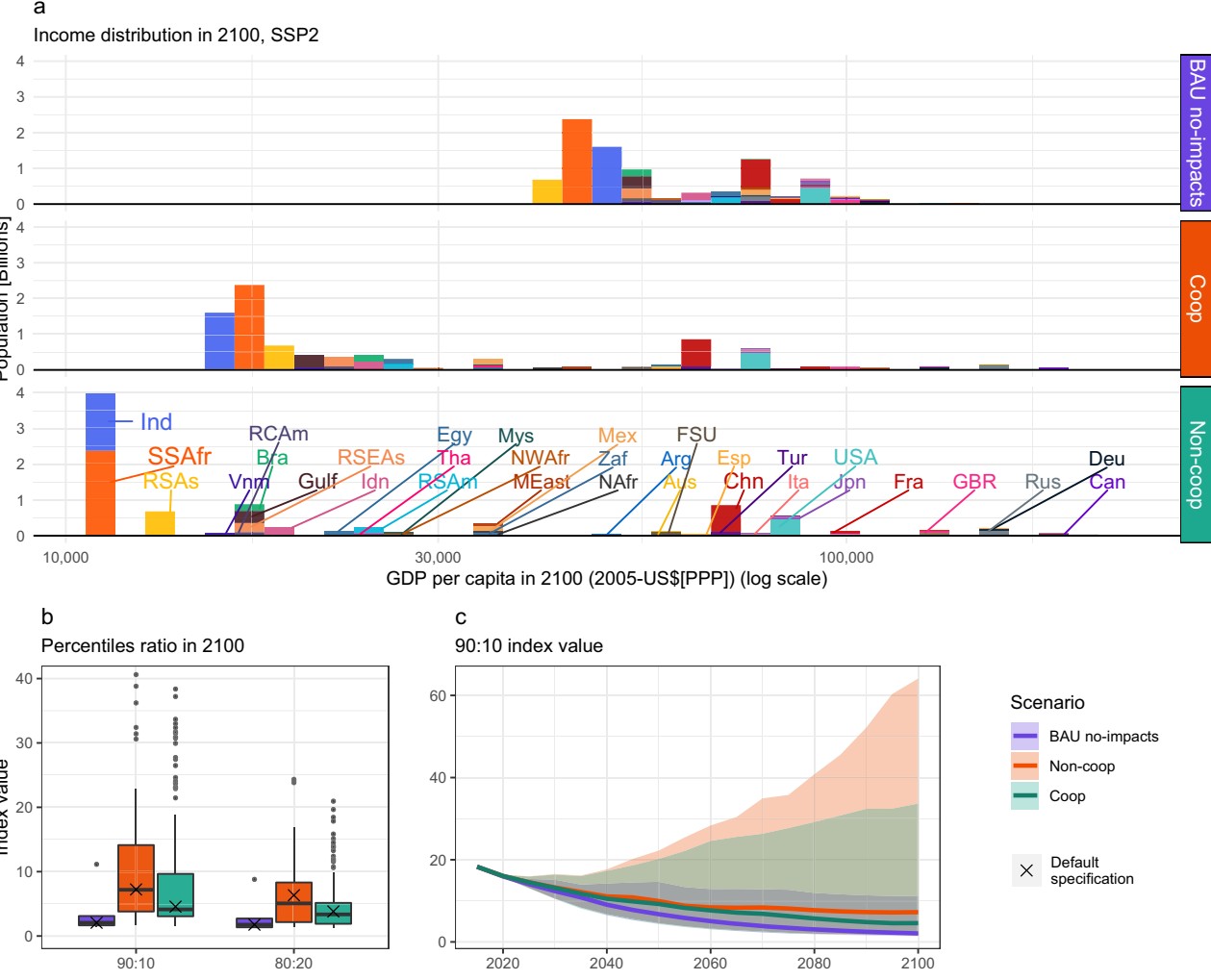

**Fig. 4 Income distribution and inequality indexes. a** GDP per-capita, net of costs and impacts, population-weighted distribution in 2100 under our default specification. **b** 90:10 and 80:20 inequality index (percentile-ratios over population-weighted distributions), in 2100, of all scenarios. Year 2020 values are 15.9 (90:10 ratio) and 3.2 (80:20 ratio). **c** 90:10 index evolution over time among all scenarios (The scenarios based on the SSP2 pathway are shown as solid lines).

climate. We provide new insights into the distributional implications for a wide array of scenarios and preferences parameterization. Regional detail allows us to systematically explore international cooperation, socioeconomic assumptions, impact function specifications, discounting assumptions, and most importantly inequality aversion. The analysis is a first, modest step towards broadening human welfare accounting into IAMs; nonetheless, the unit of analysis remains coarse compared to pressing social issues such as poverty and other forms of discrimination.

Results confirm the economic optimality of the 2 degrees goal in a regionally disaggregated setting. However, this result is obtained solely under assumptions of full cooperation and immediate action, highlighting the need of high institutional capacity to attain the Paris agreement[14]. We show that rich nations take on more stringent mitigation efforts under sufficient redistribution preferences towards poorer countries. Thus, we relate welfare analysis to burden-sharing equity principles. However, due to the geographical distribution and the persistence of climate economic impacts, climate change increases inequality. This can only be partially reduced by mitigation action. Even under optimistic assumptions about international climate agreements and how much the world cares for inequality, climate change regressive impacts persist.

These results lead to policy recommendations about time left for useful action. Besides increasing mitigation efforts and fostering cooperation, efforts should also go towards developing alternative climate policy strategies. These include $CO_2$ removal technologies, if deployed at scale and relatively fast, or, possibly, geoengineering strategies[51]. Existing assessments from detailed-process IAMs typically foresee the use of $CO_2$ removal in the second half of the century[52,53], with negative repercussions for inter-generational equity[54]. Our analysis suggests that if climate change impacts are persistent as estimated by Burke et al.[15] and others, then $CO_2$ removal should be anticipated and used earlier to complement traditional emission reduction options. Resilient socioeconomic development and adaptation planning and financing, particularly in the worst affected countries, will help manage the increased disparities brought about by climate change. Besides acting on emission reductions, mechanisms to compensate climate-induced inequalities and promote inclusive socio-economic development are needed.

Finally, it is worth mentioning the potential limitations of empirical impact functions over long timescales. They are based on historical observations and may misrepresent future economic responses and adaptation to temperature variability (see also[55]). Moreover, many relevant dimensions of heterogeneity are not

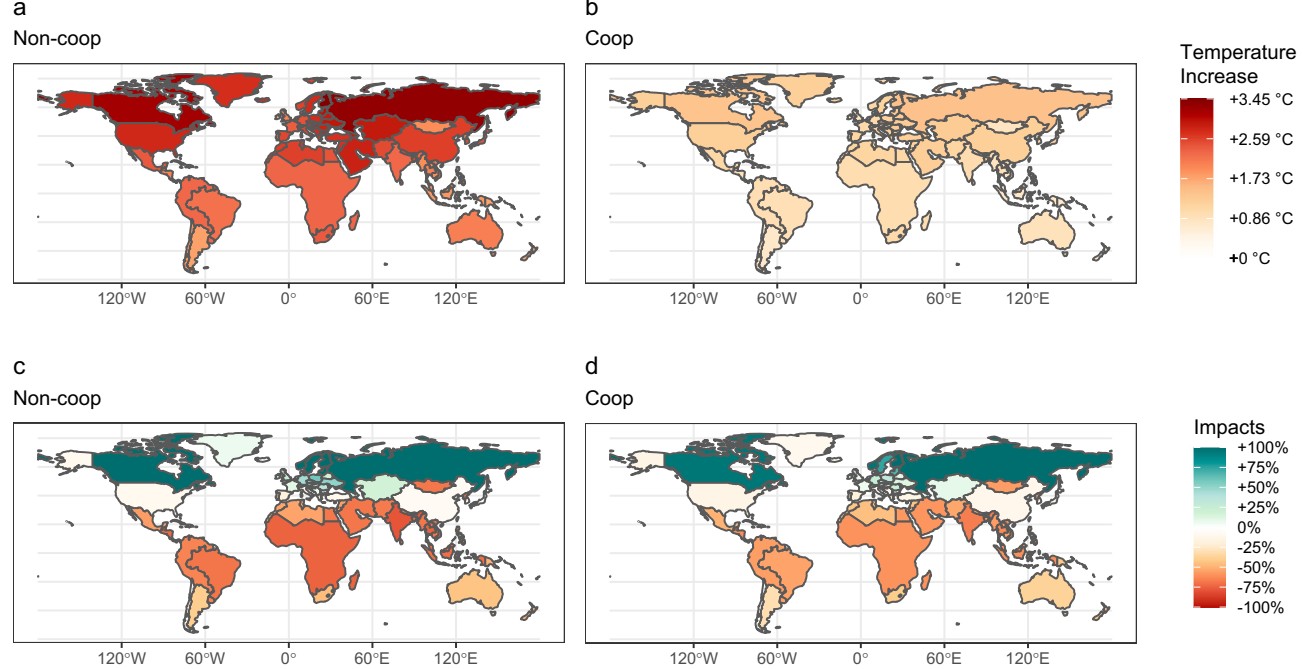

**Fig. 5 Map of population-weighted temperature increases and climate impacts projected in 2100. a, b** Local temperature increases are compared to the 1980–2010 average. **c, d** Impacts [% of GDP] projected for the year 2100. All scenarios use the default specification.

represented within this relatively simple model. These include socioeconomic impacts and sectoral detail on the energy system, land-use changes and agriculture, and biodiversity and nature. While many of these impacts have been implemented in process-based IAMs, other dimensions are still absent from the analysis. These include multifaceted inequalities (e.g., race, basic human needs, gender, see[56]), interactions with environmental risks (such as health), adaptation (such as adaptive capacity) and mitigation efforts (e.g., due to unequal patterns of consumption). The current paper has shown that socially relevant theoretical frameworks can be operationalized and can highlight the need for novel solutions. More work is needed to incorporate inequality in model-based assessment of climate change.

## Methods

**Regional aggregation.** Regions in RICE50+ model are based on the finest regional disaggregation of the POLES-EnerData abatement costs data. We kept European Union and G20 countries with country-level decision making, while others are grouped into larger macro regions. These include gulf Arabic states (golf57), former Soviet-Union (ris), middle-East group (osea), and minor countries from South-East-Asia (rsas), Latin-America (rsam), and Sub-Saharan Africa (rsaf).

**Economy.** RICE50+ largely inherits from the Nordhaus DICE model its economic representation. GDP output is computed, for each region $i$, via a Cobb-Douglas production function of capital $K_i(t)$ and labor $L_i(t)$, with total factor productivity $\text{TFP}_i(t)$:

$$Y_{\text{GROSS},i}(t) = \text{TFP}_i(t) \cdot K_i(t)^\alpha \cdot L_i(t)^{1-\alpha}. \quad (1)$$

We consider labor and TFP projections exogenous, and calibrate them to match, in the BAU case, the Shared Socioeconomic Pathways (SSPs) population and GDP pathways[34]. Since SSPs data cover 2015–2100 period only, beyond 2100 we extended the projections by linearly extrapolating the last growth rate, progressively reducing it towards zero in 2200 so that from 2200 GDP and population stabilize.

In DICE/RICE models, savings rates $S_i(t)$ are usually left as free variables to be optimized. They determine investments and capital accumulation according to equations:

$$I_i(t) = S_i(t) \cdot Y_i(t) \quad (2)$$

and:

$$K_i(t+1) = (1-\delta_k)^{\Delta t} \cdot K_i(t) + \Delta t \cdot I_i(t). \quad (3)$$

Variable $Y_i(t)$ is the final GDP resulting after subtracting abatement costs $\Lambda_i(t, \mu_i)$ to GDP net-of-damages $Y_{\text{NET},i}(t)$:

$$Y_i(t) = Y_{\text{NET},i}(t) - \Lambda_i(t, \mu_i). \quad (4)$$

The optimization of savings rates wasn't affecting the results in a relevant way for the present analysis, while it substantially increased model complexity and computational time. Moreover, when optimizing with endogenous savings rate, the model must include a simpler approximation (Eq. (19)) for impacts (described later in impacts section). Hence, we opted for fixing the savings rates, starting from historical values and linearly converging to the DICE-2016R2 long-term projection $\overline{S}$ by the year 2200. In Supplementary Fig. 11 endogenous and fixed savings rate results are compared for representative scenarios. Panel a and b show how endogenous savings lead to slightly lower global emissions and temperatures. However, panel c, confirms inequality results across all SSPs scenarios (with a noteworthy exacerbation for endogenous savings in SSP4 -persistent inequality pathway- deciles indexes). Thus, the main results about inequality are confirmed or exacerbated when endogenizing saving rates.

**Emissions calibration.** Baseline industrial emissions are directly related to economies output through the exogenous carbon-intensity $\sigma_i(t)$, expressing fossil-fuel-shares in economic production:

$$E_{\text{IND},i}(t) = \sigma_i(t) \cdot Y_{\text{GROSS},i}(t) \cdot \big(1-\mu_i(t)\big). \quad (5)$$

To calibrate it, we followed a two-step process. First, we calibrated SSP2 projections, starting from the DICE dynamics:

$$\sigma_i(t+1) = \sigma_i(t) \cdot \exp(g_i(t) \cdot \Delta t). \quad (6)$$

We imposed 2015 historical levels, and estimated $\overline{g_i}(t)$ values that minimize the difference between resulting emissions, EnerData baselines (available for 2025–2040 period), and regional levels from SSP2-marker-model (Message-GLOBIOM).

Beyond 2100 we opted for a smooth convergence, for each region, to DICE-2016R2 global carbon intensity levels by year 2200. At each point in time carbon intensity is the result of a convex-combination of two components:

$$\sigma_{ssp2,i}(t) = (1-\text{cc}(t)) \cdot \overline{\sigma_i}(t) + \text{cc}(t) \cdot \sigma_{\text{DICE}}(t), \quad (7)$$

with coefficient $\text{cc}(t)$ following a smooth sigmoid transition from 0 to 1 for $t \in [2100, 2200]$.

Then, we evaluated carbon intensities for the other SSPs by including an SSP-dependent, region-independent, multiplier $m_i(ssp)$ in Eq. (6), including previously-optimized $\overline{g_i}(t)$ term:

$$\sigma_i(ssp, t+1) = m_i(ssp) \cdot \sigma_i(ssp, t) \cdot \exp(\overline{g_i}(t) \cdot \Delta t). \quad (8)$$

As before, we fixed 2015 levels and computed $\hat{m}(ssp)$ values that minimize differences between projected emissions and regional emissions for each SSP-

Marker model. Beyond 2100 we kept the convex combination between calibrated curves and DICE global carbon intensity.

**Abatement costs**. Regions optimize the fraction of baseline emissions to abate, $\mu_i(t)$, in the range $[0, 1.2]$, with an abatement greater than 1 corresponding to negative emissions. We evaluated region-specific abatement costs curves starting from the Enerdata dataset, which includes industrial $CO_2$-abatement levels for several carbon prices over the 2025–2040 time period. We fitted the data to find the best representative continuous MAC curve. Comparing R-squared goodness-of-fit measures, we selected the best candidate, a fourth-exponent polynomial curve:

$$MAC_i(t, \mu) = a_i(t)\mu_i + b_i(t)\mu_i^4. \quad (9)$$

After calibrating region-specific coefficients, we added a multiplier correction-factor $v(t)$, equal for each region, to better match these curves to the state-of-the-art assumptions in the Integrated Assessment Modelling community. We used policy scenarios with carbon prices projections from SSPs database to extract several MAC curves. We used those curves to find the best value $\overline{v}(t)$, which minimizes the difference between RICE50+ global emissions abated and the SSP ensemble's median levels.

After 2100, we extend last data-fitted regional curves, and use same correction-factor $v(t)$ to lead a progressive convergence towards a common backstop-technology curve $BT(t)$, which follows the original DICE-2016R2 definition. The transition phase begins after 2040 and lasts until time $t_{BT}$, when each region starts matching the original backstop values for a 100% mitigation level $\hat{\mu}_i$:

$$v(t) \cdot (a_i(t)\hat{\mu}_i + b_i(t)\hat{\mu}_i^4) = BT(t)|_{t \geq t_{BT}}. \quad (10)$$

Regional total mitigation costs $\Lambda$ are then linked to its mitigation level $\mu_i(t)$ according to the equation:

$$\Lambda_i(t, \mu_i) = \int_0^{\mu_i} E_{BAU,i}(t) \cdot MAC_i(t, \mu_i) d\mu \quad (11)$$

and therefore:

$$\Lambda_i(t, \mu_i) = v(t) \cdot E_{BAU,i}(t) \cdot \left( \frac{a_i(t)}{2}\mu_i^2 + \frac{b_i(t)}{5}\mu_i^5 \right) \quad (12)$$

Differently from the original DICE formulation, we constrain the maximum mitigation increase rate over time to reflect inertia in mitigation technologies. Following Hänsel et al.[12] we set a 20% maximum increase every 5-years period:

$$\mu_i(t+1) \leq \mu_i(t) + 0.2. \quad (13)$$

**Global and local climate**. The concentration of greenhouse gases is modeled through a three-box carbon sink model. Radiative forcing $RF(t)$ is computed based on the changes in its concentration $M_{CO_2}(t)$ from pre-industrial levels $M_{CO_2,pre}$, and the exogenous addition of Other-GHG contribution:

$$RF(t) = \alpha \times \left( \ln\left(M_{CO_2}(t)\right) - \ln\left(M_{CO_2,pre}\right) \right) + RF_{OGHG}(t) \quad (14)$$

with $\alpha = 5.35$. The global atmospheric temperature increase $\Delta GMT(t)$ is computed following the DICE-2016R2 two-layer model, re-calibrated in its exchange-coefficients to match the MAGICC6 behavior[35].

To perform the impact evaluation at the country or regional level at great detail, regional temperature responses are also needed to properly consider the significant heterogeneous warming response. To this end, we implemented a statistical downscaling method based on the CMIP5 database[36]. It provides historical data and projections of temperature and precipitation at the 0.5° gridded level. We aggregated values to the country and year average level using population weights, obtaining data for $N = 244$ countries and territories. Finally, we used the global temperature data from the different representative concentration pathways (RCPs), implemented by several global climate models. We considered the median of the model ensemble to link global mean temperature increase ($\Delta GMT$) to the country-level average annual temperature for all the RCPs.

Based on this data set, we run a linear regression to estimate the local temperature levels in region $i$ at time $t$ (denoted as $T_i(t)$ and measured in °C) as a consequence of global temperature increase $\Delta GMT(t)$:

$$T_i(t) = \alpha_i + \beta_i \Delta GMT(t) \quad (15)$$

The $R^2$ of the estimated regressions varies between 0.95 and 0.999. Finally, we aggregated country-level estimates to get the values $(\alpha_i, \beta_i)$ for the 57 model regions.

**Implementation of growth impacts**. While the original DICE and other similar IAMs implement damages based on the level of GDP per period, we implemented different empirically calibrated specifications (*spec*) of linear impacts on the per-capita growth rate $\delta_{i,spec}(t)$. This factor is then applied to the GDP per-capita

growth rate $g_i(t) = \frac{Y_{NET,i}(t)}{L_i(t)} / \frac{Y_{NET,i}(t-1)}{L_i(t-1)} - 1$:

$$\frac{Y_{NET,i}(t)}{L_i(t)} = \frac{Y_{NET,i}(t-1)}{L_i(t-1)}(1 + g_i(t) + \delta_{i,spec}(t)). \quad (16)$$

By combining this impact function specification with the traditional impact definition in DICE given by

$$Y_{NET,i}(t) = \frac{Y_{GROSS,i}(t)}{\Omega_i(t)}, \quad (17)$$

and Eqs. (1), (2), and (3), we obtained a new recursive formula for impacts $\Omega_i(t)$:

$$\Omega_i(t+1) = \frac{TFP_i(t+1)}{TFP_i(t)} \left( \frac{L_i(t+1)}{L_i(t)} \right)^{-\alpha} \cdot \Upsilon_i(t)^\alpha \cdot \frac{1 + \Omega_i(t)}{(1 + g_i(t) + \delta_{i,spec}(t))^{\Delta t}} - 1, \quad (18)$$

where:

$$\Upsilon_i(t) = (1 + \delta_k)^{\Delta t} + \Delta t \cdot S_i(t) \cdot TFP_i(t) \cdot \left( \frac{L_i(t)}{K_i(t)} \right)^{1-\alpha} \cdot \frac{1}{1 + \Omega_i(t)}.$$

While this implementation is perfectly consistent with the growth-rate empirical estimation, it can lead to numerical issues, in particular when the savings rate is endogenous. Therefore, in this case, we implemented also an approximate equivalent rule to the standard $\Omega_i(t)$ as in DICE. In an economic growth model with a Cobb-Douglas production function, stable capital-labor ratios, and "small" exogenous annualized growth rates $g_{it}$, the Burke et al.[15] or similar damage function based on temperature-dependent annual growth impacts $\delta_{it}$ is approximately equivalent to the following recursive formula:

$$\tilde{\Omega}_i(t+1) = (1 + \tilde{\Omega}_{it}) \frac{1}{(1 + \delta_{i,spec}(t))^{\Delta t}} - 1. \quad (19)$$

The analytical proof and further discussion are reported in[57].

Lastly, cumulative growth impacts can lead to very high positive or negative impacts over three centuries for some small countries. To avoid this degenerating trend and the risk of biasing model optimal decisions, we limited GDP impacts within a $[+100\%, -100\%)$ interval with respect to the baseline.

**Burke et al. (2015) impact function**. The regional temperature patterns allowed us to integrate an impact function based on Burke et al.[15]. Using long-run estimates and a single equation for all countries, the authors obtained a function of growth effects directly related to country-level temperature $T_i(t)$:

$$h(T_i(t)) = 0.0127 \cdot T_i(t) - 0.0005 \cdot T_i(t)^2. \quad (20)$$

Impacts $\delta_{i,BHM}(t)$ on the production growth rate are computed as the difference between the value of this function at time $t$ and its value at the reference average temperature between 1980 and 2010 $T_{i0}$:

$$\delta_{i,BHM}(t) = h(T_i(t)) - h(T_{i0}). \quad (21)$$

**Dell, Jones and Olken (2012) impact function**. In the paper by Dell et al.[17], another linear relationship between temperature and economic growth is estimated. The parameter $\delta_{i,DJO}(t)$ yields the following main specification based on a (almost insignificant) general effect, and a strong negative effect of about additional 1.655 percentage point reductions in growth for poor countries (i.e., having GDP per-capita [PPP] below the median in the base year):

$$\delta_{i,DJO}(t) = 0.00261 \cdot (T_i(t) - T_{i0}) - 0.01655 \cdot (T_i(t) - T_{i0}) \mathbb{1}_{GDP_{CAP,i}(t_0) < Median(GDP_{CAP,i}(t_0))} \quad (22)$$

**Kahn et al. (2019) impact function**. A third empirical paper by Kahn et al.[18] similarly estimates a linear relationship, but differentiated for increases and decreases of the country-level temperatures over the historical norm. Using their main specification with $n = 30$ years for computing the historical norm as moving average (in our case starting from 1980–2010 in consistency with the case of Burke et al.[15]), we obtain a third specification for the growth effect $\delta_{i,Kahn}(t)$. Their main results conclude that a temperature increase by one degree over the historical norm is associated with a growth rate reduction by 5.86 percentage points. In comparison, a decrease by one degree implies a reduction of growth by 5.20 percentage points. The authors don't find significant difference between rich and poor countries.

$$\delta_{i,Kahn}(t) = -0.0586\left( [T_i(t) - \overline{T}_i(t-1)] - [T_i(t-1) - \overline{T}_i(t-2)] \right) \mathbb{1}_{T_i(t) > \overline{T}_i(t-1)}$$
$$-0.0520\left( [T_i(t) - \overline{T}_i(t-1)] - [T_i(t-1) - \overline{T}_i(t-2)] \right) \mathbb{1}_{T_i(t) < \overline{T}_i(t-1)} \quad (23)$$

with $\overline{T}_i(t-1) = n^{-1} \sum_{\tau=1}^{n} T_i(t-\tau)$ for $n = 6$ (each t accounts for 5 years).

**Welfare**. In the original RICE model, the social welfare function is specified as follows:

$$W = \sum_i \sum_t w_i(t) \cdot L_i(t) \cdot \left( \frac{(C_{CAP,i}(t))^{1-\eta} - 1}{1 - \eta} - 1 \right) \cdot (1 + \rho)^{-t} \quad (24)$$

While Negishi weights $w_i(t)$ have been used in the regional RICE model, their distortion of inter-temporal preferences has been criticized and their welfare economic implications are at odds with welfare economics (see, e.g., Stanton[58], Dennig and Emmerling[59]). We, therefore, implemented an alternative welfare function that has as special cases the standard welfare function, while on the other end replicating the idea of simply maximizing global consumption (as proposed in Stanton[58] as one solution), and allows a gradual change from equal marginal utility to population weighting. This is implemented through an additional parameter of inequality aversion $\gamma$ in the welfare specification (see also Berger and Emmerling[60]):

$$W = \sum_{t=1}^{T} \left[ \frac{1}{1-\eta} \left( \sum_i w_{pop,i}(t) \left( \frac{C_i(t)}{L_i(t)} \right)^{1-\gamma} \right)^{\frac{1-\eta}{1-\gamma}} - 1 \right] \cdot (1 + \rho)^{-t} \quad (25)$$

with population-weights $w_{pop,i}(t) = L_i(t)/(\sum_i L_i(t))$ and $\gamma \neq 1$ condition. Here, $\rho$ denotes the utility discount rate, set to 1.5% in our default specification equal across regions and over time, while, as in DICE, we use $\eta = 1.45$ as inverse of the inter-temporal elasticity of substitution, which is close to what an expert elicitation on this parameter has found[61]. For $\gamma = 0$, the objective becomes to maximize world average consumption while for $\gamma = \eta$, the formulation collapses to the standard DICE welfare function. For the value of $\gamma$, Atkinson and Brandolini[40] consider values between 0.2 and 2.5 as defensible. For our default specification we chose an intermediate value of $\gamma = 0.5$, e.g., close to the value found in[41] or values used in[42]).

**Other GHGs and Land-Use**. We kept Land-Use (LU) and other greenhouse gases (OGHG) as exogenous addition to the main model dynamics. For Land-Use, we retrieved regional starting levels $E_{LU,i}(t_0)$ from the country-level PRIMAP-hist database[62]. We took the mean values between 2010 and 2015 to average out historical fluctuations, common in LU emissions. We kept original DICE-2016R2 decreasing trend and differentiated between two alternative cases. In the first one, used for BAU scenarios, all countries are affected by the decreasing trend. This will lead high-emitting countries to lower their emissions over time and countries that already start from negative values to increase their emissions towards the common zero-value asymptote. In the second case, used for benefit-cost scenarios, we apply this reduction only to those countries that start from positive values, keeping constant negative ones.

For the OGHGs, we retrieved data from SSPs models comparing relative forcing contribution $RF_{OGHG}$ to the forcing from $CO_2$ ($RF_{CO_2}$). Given that we found a relatively close linear relationship ($R^2 = 0.608$) between both global forcing contributions and across all scenarios, we added OGHGs using the regression-estimated parameters (slope = 0.199, intercept = −0.011) in the model.

## Data availability
All data generated and used in this analysis can be accessed at https://github.com/witch-team/RICE50xmodel/releases/download/v1.0.0/NC2021_results_dataset.zip.

## Code availability
All code generated and used in this analysis can be accessed at https://github.com/witch-team/RICE50xmodel.

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

## Acknowledgements

The research leading to these results has received funding from the European Union's Horizon 2020 research and innovation programme under grant agreement no. 821124 (NAVIGATE) and No. 776479 (COACCH). The usual caveat applies.

## Author contributions

M.T. conceived the study. P.G. developed the model, with support from all authors. M.T., P.G. and J.E. drafted the article. P.G., J.E., G.M., A.C., K.I.W., A.H., M.T. contributed to the final version.

## Competing interests

The authors declare no competing interests.
