## [Peer Review File · Nature Communications]

REVIEWER COMMENTS

Reviewer #1 (Remarks to the Author):

This work develops a regional distinguished global IAM using econometric estimated damage functions. The novelty of this paper is that it applies different mitigation scenarios in combination with uncertainty over key parameters, inequality aversion and recent econometric impacts estimates. All these elements have been implemented in the literature, but not simultaneously. Particularly the mitigation scenarios and damage functions are the core contribution of this paper in my view. As the authors mention, global mitigation IAMs have been updated to include the BHM damage estimates. The novelty of this work then would be the regional dimension then. The paper is written well, the figures are clear and the analysis itself interesting. The methods are well developed. I believe that the work itself is relevant for this publication, however, certain revisions would be needed in my view.

My comments regarding this work include the framing of the work (which is one of my main concerns with this work), some technical issues and some minor issues.

Framing and literature review

The main conclusion of this paper (that climate change increases inequality even with global cooperation) is presented as being novel. It is not. The same conclusions have been drawn from many IAMs starting with the RICE model (Nordhaus and Boyer 2000), the AD-RICE model (de Bruin et al. 2009) and recently in other literature (Taconet et al., 2020).

Going beyond this, I would say that the conclusion is obvious given that climate change impacts are higher in developing (poorer) regions. This is illustrated in the paper in figure 3, which is a really great figure. Impacts are higher as GDP is lower. So for there not to be higher inequality with climate change, mitigation costs need to be higher than damage costs and be born by richer regions. Given an economic optimum ($MC=MB$) this would not be the case and mitigation would fall short of reaching the level of impacts. In summary, the fact that impacts are higher for poorer regions is not new and leads in itself to the conclusion that cooperation does not undo the inequality increase of climate change.

I feel the authors do not give a complete overview of the literature. Other IAMs are dismissed as being outdated and not containing the latest climate and economic evidence. I agree concerning the climate evidence, but I do not agree concerning the economic evidence. Econometric estimates of impacts such as BHM and other impact estimates apply completely different methodologies. Both methods has their advantages and this should be portrayed correctly. The authors do not discuss the limitations of econometric estimates of climate change impacts. Given the limited climate change in the past, one can question the validity of such estimates in projecting future impacts. Furthermore, there is a limited understanding of what drives these impacts. To give an example in developing countries short term and historical impacts from climate change are dominated by health impacts. Health impacts however are extremely dependent on GDP growth and are expected to play a much smaller role in the future (see e.g. de Bruin and Ayuba 2020). Such a mechanism would not be captured in econometric estimates. In this regard, I believe that this work is missing a key reference to the OECD ENV linkages model (Dellink et al 2014.), which includes one of the most detailed modelling analysis of climate change impacts. Further, Dellink et al. e.g. include details on the channels through which climate change impacts work. Even RICE, AD-RICE, AD-MERGE and FUND have this advantage of sector level impacts (and the added advantage of the inclusion of adaptation which is lacking here). Again I do not think that these econometric estimates should not be applied, but a sentence or 2 mentioning the critique of these estimates or the advantages of the other damage assessments would be necessary. I believe that given the background of the readers of this publication a fair description of the literature is necessary.

Technical comments

On page 16 the authors state:

For this paper, the optimization of savings rates wasn't affecting the analysis in a relevant way, while it substantially increased model complexity and computational time. Hence, we fixed the savings rates, starting from historical values and linearly converging to the DICE-2016R2 long-term projection S by the year 2200.

Though I agree that when running different mitigation scenarios, the optimal savings rate does not differ much, when changing parameter values, significant differences are found. A quick run of the AD-DICE2016 model shows that e.g. choosing a 50% higher initial TFP growth rate leads to an optimal savings rate of half the base savings rate in period 5 and one 10 times lower in period 9. It seems highly unlikely to me that in all the scenarios examined in this work that fixing the savings rate does not impact results. Have the authors checked for this?

In this work impacts are modelled as a function of Ynet and not Ygross (as in DICE). This has significant impact on results specifically when impacts are high. Have the authors considered this and can they justify the use of Ynet as opposed to Ygross?

Concerning the analysis of cooperation and inequality aversion. I am not familiar with the method applied here, so if I have misunderstood, excuse me. As I understand in the cooperative solution total global welfare is maximised. Which would mean that mitigation is undertaken as a form of both reducing climate change to the optimum level (where the global marginal costs of mitigation equal the marginal global benefits). And to basically transfer money to other regions and hence increase global welfare as mitigation can be used to reduce impacts in poorer regions at the cost of richer regions and hence increase global welfare (see de Bruin 2009). Is this the case or does the method ensure the global mitigation optimum is found? If yes, how?

Minor comments:

The specifications span degrees of adaptation responses. Page 3

I do not understand what is meant by this.

On figure B5a, I cant read BAU no impacts line

Page 5: This is consistent with mitigation efforts in the major world economies consistent with climate stabilization [47].

Something is wrong here..

Page 7:

Thus, residual climate damages remain highly regressive independently of equity preferences; these mostly affect mitigation efforts and costs.

Also elsewhere (e.g. figure 5)

The use of the word residual damages is confusing. Residual damages refers to damages after adaptation. I do not think this analysis includes adaptation hence residual damages is the wrong term.

Page 8

Emissions historically and in The coming decades are sufficient to irreversibly increase inequality between countries and lead to strong residual climate impacts even if global cooperation is achieved, as shown in Fig.5

The?

References

de Bruin, K.C., R.B. Dellink, R.S.J. Tol (2009). 'International cooperation on climate change adaptation from an economic perspective', ESRI Working Paper 323, Dublin.

de Bruin, K.C, and Ayuba, V. (2020). 'What does Paris mean for Africa: an Integrated assessment', ESRI Working Paper 633, Dublin.

Dellink, R., et al. (2014), "Consequences of Climate Change Damages for Economic Growth: A Dynamic Quantitative Assessment", OECD Economics Department Working Papers, No. 1135, OECD

Publishing, Paris, <https://doi.org/10.1787/5jz2bxb8kmf3-en>.

Taconet et al: <https://link.springer.com/article/10.1007/s10584-019-02637-w>

Overall, I thoroughly enjoyed this paper,

Kind regards

Kelly de Bruin

Reviewer #2 (Remarks to the Author):

At a technical level, this paper makes a very large contribution to the "climate economics" literature by refining the traditional aggregate cost-benefit calculations that have been done for 3-4 decades to include some important equity dimensions. As such, I think it would be quickly embraced by the 1-2% of the world that thinks welfare economics as currently practiced is a useful if not essential way to think about climate policy. I think what is missing is a more complete admission of the challenges yet remaining to make this kind of work to the other 98-99% of the world. I think this can be at least partially remedied by re-framing what has been done and caveating the results in a way to which these parties can relate without requiring additional analysis.

In many ways the paper does a good job of showing some of the shortcomings of doing cost benefit analysis of climate change that aggregates and averages across large groups of people in a way that ignores or distorts what the stakes might be for smaller groups within these aggregates, but does not go far enough in describing the significant challenges that remain—details that are likely more important to most citizens, and national and international climate negotiators. In a way this will read to many of them as an attempt to show that the welfare economics framework as currently conceived is useful and can be made more directly useful with a group of technical refinements and implicitly we see you and feel you all, so if you give us a few more decades to refine our "empirical" estimates we will provide you some useful information to help you decide what to do. Thus, we have less aggregated regions and income distribution aware dynamic damage functions, but no real detail on sectoral and regional impacts where the impacts of climate change and climate change impacts are likely to be felt most acutely. Thus, human behavior and welfare is considered more broadly than before, but the people who are most severely impacted have no faces and are generally left out and/or implicitly undervalued or not valued at all. And we end up with some appropriate caveats like the following statement in the conclusions section not being revealed until that point: "Results confirm the economic optimality of the 2 degrees goal in a regionally dis-aggregated setting. However, this result is obtained solely under assumptions of full cooperation and immediate action, highlighting the need of high institutional capacity to attain the Paris agreement." Thus, we are left in the end with a result that is valid only under extremely unrealistic assumptions without knowing the full range of scenarios that could emerge if these assumptions are not satisfied. Here the paper does go a lot further with consideration of a range of SSP scenarios and "non-cooperative" results, and results that do consider "income distribution" as a general criteria, rather than a personal level. There are many examples of where this falls short of being useful to many stakeholders, but for one obvious example consider a poor low lying area where sea level rise threatens to wipe out your heritage and very way of life by submerging it all under water for a long long time.

Thus, I would definitely favor publication of this paper as a very real and important contribution to the economics of climate change, but not without a better framing and set of caveats that talk not only about what is now included and why it is important, but a large number of things that are still left out. Otherwise, I think it will be perceived as more of the same old largely irrelevant and probably misleading results, that claims to address welfare without clarifying whose welfare is being maximized, how and why. Put differently, it is hard to address welfare even at this level of aggregation without a larger dose of humility and empathy for fate of the non-elites among us.

Authors response to referee

Nature Communications manuscript
NCOMMS-20-44598-T

Reviewer 1

(Remarks to the Author):

This work develops a regional distinguished global IAM using econometric estimated damage functions. The novelty of this paper is that it applies different mitigation scenarios in combination with uncertainty over key parameters, inequality aversion and recent econometric impacts estimates. All these elements have been implemented in the literature, but not simultaneously. Particularly the mitigation scenarios and damage functions are the core contribution of this paper in my view. As the authors mention, global mitigation IAMs have been updated to include the BHM damage estimates. The novelty of this work then would be the regional dimension then. The paper is written well, the figures are clear and the analysis itself interesting. The methods are well developed. I believe that the work itself is relevant for this publication, however, certain revisions would be needed in my view.

My comments regarding this work include the framing of the work (which is one of my main concerns with this work), some technical issues and some minor issues.

Framing and literature review

The main conclusion of this paper (that climate change increases inequality even with global cooperation) is presented as being novel. It is not. The same conclusions have been drawn from many IAMs starting with the RICE model (Nordhaus and Boyer 2000), the AD-RICE model (de Bruin et al. 2009) and recently in other literature (Taconet et al., 2020).

We concur with the referee that inequality and heterogeneity have been analyzed in the modeling literature. We also thank her for pointing out references such as (de Bruin et al. 2009). We now credit these previous works more in the text. We feel our paper goes beyond the available literature by combining and advancing model features such as the explicit representation of inequality, the higher regional granularity, the empirically grounded damage functions, the welfare-maximizing policy scenarios, and SSPs-calibrated projections. We also clarified better what gives novelty and differentiates ourselves in lines 30-60 of the manuscript and structured better "Implementation of growth impacts" section, lines 549-561 of Methods. We also rephrased the abstract.

Going beyond this, I would say that the conclusion is obvious given that climate change impacts are higher in developing (poorer) regions. This is illustrated in the paper in figure 3, which is a really great figure. Impacts are higher as GDP is lower. So for there not to be higher inequality with climate change,

mitigation costs need to be higher than damage costs and be born by richer regions. Given an economic optimum ($MC=MB$) this would not be the case and mitigation would fall short of reaching the level of impacts. In summary, the fact that impacts are higher for poorer regions is not new and leads in itself to the conclusion that cooperation does not undo the inequality increase of climate change. I feel the authors do not give a complete overview of the literature. Other IAMs are dismissed as being outdated and not containing the latest climate and economic evidence.

We agree that an increasing level of inequality has been already shown by several models and studies. However, our inequality quantification relies on the most recent empirical evidence, which is novel in regional benefit-cost optimizing models. To the best of our knowledge growth impacts have been introduced only in a global DICE model so far (i.e., Glanemann 2020). Those impacts indeed project highly regressive impacts which lead quite obviously to increased inequalities. But the extent to which the inequalities driven by the recent climate impact literature persist, even in welfare-maximizing pathways, is a novel result in our view. We also emphasize the major role of economic convergence as captured by different SSPs pathways.

I agree concerning the climate evidence, but I do not agree concerning the economic evidence. Econometric estimates of impacts such as BHM and other impact estimates apply completely different methodologies. Both methods has their advantages and this should be portrayed correctly. The authors do not discuss the limitations of econometric estimates of climate change impacts. Given the limited climate change in the past, one can question the validity of such estimates in projecting future impacts. Furthermore, there is a limited understanding of what drives these impacts. To give an example in developing countries short term and historical impacts from climate change are dominated by health impacts. Health impacts however are extremely dependent on GDP growth and are expected to play a much smaller role in the future (see e.g. de Bruin and Ayuba 2020). Such a mechanism would not be captured in econometric estimates.

We agree that the econometric estimates raise new important issues, and do not explain relevant mechanisms. Our results show, as evidenced in the discussion section, significant robustness to different empirically based impact specifications. Functions adopted cover diverse assumptions on economic responses, impact lags, rich/poor differentiation. However, we agree on the necessity of a better discussion on the limitations of the adopted econometric estimates. We therefore added the following closing remarks (lines 240-250, the second half addressing the concerns of referee 2):

“Finally, it is worth mentioning the potential limitations of empirical impact functions over long timescales, as they are based on historical observations and may misrepresent future economic responses and adaptation to temperature variability (see also Burke et al., 2018). Moreover, many relevant dimensions of heterogeneity are not represented within this relatively simple model. These include socio-economic impacts and sectoral detail on the energy system, land-use changes and agriculture, and biodiversity and nature. While many of these impacts have been implemented in process-based IAMs, other dimensions are still absent from the analysis. These include multifaceted inequalities (e.g. race, basic human needs, gender), interactions with environmental risks (such as health), adaptation (such as adaptive capacity) and mitigation efforts (e.g., due to unequal patterns of consumption). The

current paper has shown that socially relevant theoretical frameworks can be operationalized and can highlight the need for novel solutions. More work is needed to incorporate inequality in model-based assessment of climate change.”

In this regard, I believe that this work is missing a key reference to the OECD ENV linkages model (Dellink et al 2014.), which includes one of the most detailed modelling analysis of climate change impacts. Further, Dellink et al. e.g. include details on the channels through which climate change impacts work. Even RICE, AD-RICE, AD-MERGE and FUND have this advantage of sector level impacts (and the added advantage of the inclusion of adaptation which is lacking here). Again I do not think that these econometric estimates should not be applied, but a sentence or 2 mentioning the critique of these estimates or the advantages of the other damage assessments would be necessary. I believe that given the background of the readers of this publication a fair description of the literature is necessary.

The econometric approach accounts for aggregated economic losses without sectoral detail. Other papers like Dellink et al. (2014) have actually looked at more detailed sectoral implications but do not perform benefit-cost optimization.

We acknowledge this and therefore added this sentence and reference (lines 46-50):

“ This resolution allows only to partially capture the variation in the costs and benefits of climate action, and none of these models accounts for the latest climate and economic evidence. While other models, including Computable General Equilibrium (CGE) models such as in Dellink et al. 2004, provide also sectors and regional detail, their policy questions are in most cases evaluating prescribed policy pathways rather than full dynamic optimization.”

Technical comments

On page 16 the authors state:

For this paper, the optimization of savings rates wasn't affecting the analysis in a relevant way, while it substantially increased model complexity and computational time. Hence, we fixed the savings rates, starting from historical values and linearly converging to the DICE-2016R2 long-term projection S by the year 2200.

Though I agree that when running different mitigation scenarios, the optimal savings rate does not differ much, when changing parameter values, significant differences are found. A quick run of the AD-DICE2016 model shows that e.g. choosing a 50% higher initial TFP growth rate leads to an optimal savings rate of half the base savings rate in period 5 and one 10 times lower in period 9. It seems highly unlikely to me that in all the scenarios examined in this work that fixing the savings rate does not impact results. Have the authors checked for this?

Thank you for this comment. It is indeed a relevant point given the size of climate impacts on the economy. We agree that it needed better clarification.

Therefore, we added Figure B.11 in the SI which compares fixed-savings results with endogenous-savings ones across all SSP scenarios.

We added a paragraph in Methods (economy section, lines 482-491) which reads as follows:

“In SI Figure B11 endogenous and fixed savings rate results are compared for representative scenarios. Panel a and b show how endogenous savings lead to slightly lower global emissions and temperatures. However, panel c, confirms inequality results across all SSPs scenarios (with a noteworthy exacerbation for endogenous savings in SSP4 —persistent inequality pathway— deciles indexes). Thus, the main results about inequality are confirmed or exacerbated when endogenizing saving rates.”

In this work impacts are modelled as a function of Ynet and not Ygross. This has a significant impact on results specifically when impacts are high. Have the authors considered this and can they justify the use of Ynet as opposed to Ygross?

Thanks for the opportunity to clarify this point. Please note that in the original RICE model, indeed impacts and abatement costs were directly applied to YGROSS (Eq. A12 in Nordhaus and Yang (1996)). The much more recent DICE2016 model instead applies damages to YGROSS and costs on the resulting YNET, which is exactly what we followed here, see the original DICE-2016R2 code:

```
yneteq(t) .. YNET(t) =E= YGROSS(t) * (1-damfrac(t));  
yy(t) .. Y(t) =E= YNET(t) - ABATECOST(t);
```

However, following your question we decided to add an explicit equation addressing this point. Therefore we added eq.(4) in Methods (Economy, line 480) and the sentence:

“With $Y_i(t)$ it is represented final GDP resulting after subtracting abatement costs $\Lambda_i(t, \mu_i)$ to GDP net-of-damages $Y_{NET,i}(t)$:

$$Y_i(t) = Y_{NET,i}(t) - \Lambda_i(t, \mu_i). \quad (4)$$

Concerning the analysis of cooperation and inequality aversion. I am not familiar with the method applied here, so if I have misunderstood, excuse me. As I understand in the cooperative solution total global welfare is maximised. Which would mean that mitigation is undertaken as a form of both reducing climate change to the optimum level (where the global marginal costs of mitigation equal the marginal global benefits). And to basically transfer money to other regions and hence increase global welfare as mitigation can be used to reduce impacts in poorer regions at the cost of richer regions and hence increase global welfare (see de Bruin 2009). Is this the case or does the method ensure the global mitigation optimum is found? If yes, how?

We document the choices regarding our welfare function in the dedicated Methods section. We start from a standard Utilitarian welfare function in the cooperative scenarios, as found

in RICE or similar models. We expand it by allowing inequity aversion to vary between countries and over time, similar to Epstein-Zin preferences varying intertemporal fluctuation and risk aversion, which has also been applied to (stochastic) DICE. Notably, by setting gamma to zero, the Negishi solution of RICE is recovered as a special case. But overall, we do not allow for transfers of capital or permits, but endogenously solve the global optimum under this welfare function. Hence the global solution is indeed globally optimal from a social welfare perspective, (i.e., marginal welfare benefits and welfare costs are equalized). Note this can however result in different dollar values of marginal costs and benefits, as has been noted e.g., in (Fankhauser, Tol, and Pearce 1997) and (Anthoff and Tol 2010).

Minor comments:

Page 3

The specifications span degrees of adaptation responses.

I do not understand what is meant by this.

Rephrased (line 79):

“They [BHM impact specifications] also implicitly allow for different historical adaptation to longer-run climate.”

On figure B5a, I can't read BAU no impacts line

Figure has been updated including BAU no-impacts.

Page 5:

This is consistent with mitigation efforts in the major world economies consistent with climate stabilization [47].

Something is wrong here..

Fixed (line 136):

“This is consistent with mitigation efforts in the major world economies aiming at climate stabilization”

Page 7:

Thus, residual climate damages remain highly regressive independently of equity preferences; these mostly affect mitigation efforts and costs.

Also elsewhere (e.g. figure 5)

The use of the word residual damages is confusing. Residual damages refers to damages after adaptation. I do not think this analysis includes adaptation hence residual damages is the wrong term.

Thank you, we agree. We have therefore substituted “*residual damages*” with “*climate impacts*” in all occurrences.

Page 8

Emissions historically and in The coming decades are sufficient to irreversibly increase inequality between countries and lead to strong residual climate impacts even if global cooperation is achieved, as shown in Fig.5
The?

Better rephrasing (line 204):

“The combined effect of historical emissions and those occurring in a few coming decades is sufficient to..”

References

de Bruin, K.C., R.B. Dellink, R.S.J. Tol (2009). 'International cooperation on climate change adaptation from an economic perspective', ESRI Working Paper 323, Dublin.

de Bruin, K.C. and Ayuba, V. (2020). 'What does Paris mean for Africa: an Integrated assessment', ESRI Working Paper 633, Dublin.

Dellink, R., et al. (2014), "Consequences of Climate Change Damages for Economic Growth: A Dynamic Quantitative Assessment", OECD Economics Department Working Papers, No. 1135, OECD Publishing, Paris, <https://doi.org/10.1787/5jz2bxb8kmf3-en>.

Taconet et al: <https://link.springer.com/article/10.1007/s10584-019-02637-w>

Overall, I thoroughly enjoyed this paper,

Kind regards

Kelly de Bruin

Reviewer 2

(Remarks to the Author):

At a technical level, this paper makes a very large contribution to the "climate economics" literature by refining the traditional aggregate cost-benefit calculations that have been done for 3-4 decades to include some important equity dimensions.

Thank you very much for your positive assessment.

As such, I think it would be quickly embraced by the 1-2% of the world that thinks welfare economics as currently practiced is a useful if not essential way to think about climate policy. I think what is missing is a more complete admission of the challenges yet remaining to make this kind of work to the other 98-99% of the world.

I think this can be at least partially remedied by re-framing what has been done and caveating the results in a way to which these parties can relate without requiring additional analysis.

We agree with the referee that the literature on CBA applied to climate targets has lacked realism and deliberately avoided covering critical dimensions characterizing human and natural systems. Moreover, the literature has been rather disconnected from other disciplines. In particular, we feel that embracing a more holistic approach to move away from one social planner and "representative" world citizen is vital. This consideration motivated this project, as a way to at least consider all countries with their characteristics, starting from socio-economic trends, differentiated climate impacts, and mitigation costs.

Some of the authors of this article have recently published a perspective article claiming the need to go beyond the stylized representation of economic agents typical of contemporary assessments (Emmerling and Tavoni, Representing Inequality in Integrated Assessment Modeling of Climate Change, One Earth, 2021). The short format and methodological focus of a journal such as Nature Communication has motivated us to write an article centered around the modeling and empirical assessment, without the possibility to devote sufficient space to the referee's deep questions. In the revised version, we emphasize the need to depart from stylized CBA solely focused on efficiency arguments, and the implicit normativity characteristic of existing assessments. We frame our methodological contribution as a proof of concept that an expanded notion of welfare can be incorporated in standardly used tools, can be appropriately calibrated, and can illuminate new challenges which would be left unaddressed by the standard policy prescriptions. Changes in the framing have been made to the abstract, and throughout the paper, such as in lines 32-36, 53-54, 214-220, 236-239.

In many ways the paper does a good job of showing some of the shortcomings of doing cost benefit analysis of climate change that aggregates and averages across large groups of people in a way that ignores or distorts what the stakes might be for smaller groups within these aggregates, but does not go far enough in describing the significant challenges that remain—details that are likely more important to most citizens, and national and international climate negotiators.

In a way this will read to many of them as an attempt to show that the welfare economics framework as currently conceived is useful and can be made more directly useful with a group of technical refinements and implicitly we see you and feel you all, so if you give us a few more decades to refine our "empirical" estimates we will provide you some useful information to help you decide what to do.

Thus, we have less aggregated regions and income distribution aware dynamic damage functions, but no real detail on sectoral and regional impacts where the impacts of climate change and climate change impacts are likely to be felt most acutely.

Thus, human behavior and welfare is considered more more broadly than before, but the people who are most severely impacted have no faces and are generally left out and/or implicitly undervalued or not valued at all.

And we end up with some appropriate caveats like the following statement in the conclusions section not being revealed until that point: "*Results confirm the economic optimality of the 2 degrees goal in a regionally dis-aggregated setting. However, this result is obtained solely under assumptions of full cooperation and immediate action, highlighting the need of high institutional capacity to attain the Paris agreement.*"

Thus, we are left in the end with a result that is valid only under extremely unrealistic assumptions without knowing the full range of scenarios that could emerge if these assumptions are not satisfied. Here the paper does go a lot further with consideration of a range of SSP scenarios and "non-cooperative" results, and results that do consider "income distribution" as a general criteria, rather than a personal level. There are many examples of where this falls short of being useful to many stakeholders, but for one obvious example consider a poor low lying area where sea level rise threatens to wipe out your heritage and very way of life by submerging it all under water for a long long time.

Thus, I would definitely favor publication of this paper as a very real and important contribution to the economics of climate change, but not without a better framing and set of caveats that talk not only about what is now included and why it is important, but a large number of things that are still left out.

Otherwise, I think it will be perceived as more of the same old largely irrelevant and probably misleading results, that claims to address welfare without clarifying whose welfare is being maximized, how and why. Put differently, it is hard to address welfare even at this level of aggregation without a larger dose of humility and empathy for fate of the non-elites among us.

We warmly welcome your thoughts on the topic in general and agree! We aim indeed for relevance and think we made one step forward. But we acknowledge that many critical issues have been left out from the analysis. The current paper aims to show that methodologically it is possible to expand the welfare framework to include inequality and show that doing so leads to striking different results than those of standard CBA. But the caveats, of course, don't stop here. Notably, a better inclusion of within-country dynamics (and in particular inequalities), capturing for instance poverty, but also issues like biodiversity and nature explicitly would be necessary. We now discuss these caveats in more detail in the conclusion sections. In particular, we added the following paragraphs (lines 218-220):

“The analysis is a first, modest step towards broadening human welfare accounting into IAMs; nonetheless, the units of analysis remain coarse when compared with pressing social issues such as poverty and other forms of discrimination.”

and (lines 242-250):

“Moreover, many relevant dimensions of climate change are not represented within this relatively simple model. These include socio-economic impacts and sectoral detail on the energy system, land-use changes and agriculture, and biodiversity and nature. While many of these impacts have been implemented in process-based IAMs, other dimensions are still absent from the analysis. These include multifaceted inequalities (e.g. race, basic human needs, gender), interactions with environmental risks (such as health), adaptation (such as adaptive capacity) and mitigation efforts (e.g., due to unequal patterns of consumption). The current paper has shown that socially relevant theoretical frameworks can be operationalized and can highlight the need for novel solutions. More work is needed to incorporate inequality in model-based assessment of climate change ”

References:

Anthoff, David, and Richard S.J. Tol. 2010. “On International Equity Weights and National Decision Making on Climate Change.” *Journal of Environmental Economics and Management* 60 (1): 14–20. <https://doi.org/10.1016/j.jeem.2010.04.002>.

Fankhauser, Samuel, Richard S.J. Tol, and David W. Pearce. 1997. “The Aggregation of Climate Change Damages: A Welfare Theoretic Approach.” *Environmental and Resource Economics* 10 (3): 249–66. <https://doi.org/10.1023/A:1026420425961>.